# Chemotherapy-Induced Changes in the Lung Microenvironment: The Role of MMP-2 in Facilitating Intravascular Arrest of Breast Cancer Cells

**DOI:** 10.3390/ijms221910280

**Published:** 2021-09-24

**Authors:** Justin D. Middleton, Subhakeertana Sivakumar, Tsonwin Hai

**Affiliations:** 1Department of Biological Chemistry and Pharmacology, College of Medicine, Ohio State University, Columbus, OH 43210, USA; middleton.124@osu.edu (J.D.M.); sivakumar.35@buckeyemail.osu.edu (S.S.); 2Molecular, Cellular, and Developmental Biology Program, Ohio State University, Columbus, OH 43210, USA

**Keywords:** chemotherapy, breast cancer metastasis, vascular microenvironment, cancer cell adhesion, basement membrane, matrix metalloprotease 2 (MMP-2), laminin, integrin β1

## Abstract

Previously, we showed that mice treated with cyclophosphamide (CTX) 4 days before intravenous injection of breast cancer cells had more cancer cells in the lung at 3 h after cancer injection than control counterparts without CTX. At 4 days after its injection, CTX is already excreted from the mice, allowing this pre-treatment design to reveal how CTX may modify the lung environment to indirectly affect cancer cells. In this study, we tested the hypothesis that the increase in cancer cell abundance at 3 h by CTX is due to an increase in the adhesiveness of vascular wall for cancer cells. Our data from protein array analysis and inhibition approach combined with in vitro and in vivo assays support the following two-prong mechanism. (1) CTX increases vascular permeability, resulting in the exposure of the basement membrane (BM). (2) CTX increases the level of matrix metalloproteinase-2 (MMP-2) in mouse serum, which remodels the BM and is functionally important for CTX to increase cancer abundance at this early stage. The combined effect of these two processes is the increased accessibility of critical protein domains in the BM, resulting in higher vascular adhesiveness for cancer cells to adhere. The critical protein domains in the vascular microenvironment are RGD and YISGR domains, whose known binding partners on cancer cells are integrin dimers and laminin receptor, respectively.

## 1. Introduction

Female breast cancer has surpassed lung cancer as the most commonly diagnosed new cases worldwide [1] and is a leading cause of death for women. In 2020, new cases of diagnosis were estimated to be ~2.3 million and the number of deaths around 685,000 [1]. Since metastasis is the major cause of cancer death [2], it is important to understand the mechanisms behind metastasis. The metastasis cascade can be arbitrarily divided into several steps [3]: cancer cells break through (invade) the basement membrane, migrate through the tissue parenchyma, intravasate (enter the blood vessel), survive in the blood stream, and colonize the secondary site.

Previously, we and others showed that chemotherapy, the very treatment to combat cancer, paradoxically enhances metastasis—indirectly—by modifying the non-cancer cells in the host, which in turn affects cancer cells [4,5,6]. Along the metastatic cascade, chemotherapy was shown to facilitate intravasation and colonization steps. (a) For intravasation, paclitaxel (PTX) was shown to increase the number of a micro-anatomical structures, called tumor microenvironment of metastasis (TMEM), in the primary tumor [5,6], a structure composed of a macrophage and a cancer cell in close proximity on a blood vessel [7]. TMEM is a site for cancer cells to enter the blood stream [8]. Consistently, its increase in abundance is accompanied by an increase in the circulating cancer cells [5,6]. (b) For colonization—a rate-limiting step in metastasis [9]—several chemotherapeutic drugs (PTX, cyclophosphamide (CTX), and cisplatin) have been shown to modify the host environment to make the secondary site more hospitable for cancer cells to colonize [5,10,11,12]. The experimental model used in these studies is a pre-treatment design, where mice (the host organisms) were pre-treated with chemotherapeutic drugs several days before injecting cancer cells into the blood stream: intravenous for lung colonization and intracardiac for bone colonization. This pre-treatment design allows the drugs to be excreted from the mice, thus avoiding their direct cytotoxicity to cancer cells. Consequently, any effect of the drug on the cancer burden is through an indirect path: affecting the non-cancer host cells, which in turn affect cancer cells.

The mechanism for this indirect path is not well understood. To investigate this intriguing yet devastating phenomenon, we carried out a pre-treatment experiment (described above) followed by analysis of the cancer burden (in the lung) at different time points: from 3 h to 3 days and beyond. We identified two distinct stages: (i) an early-stage at 3 h after cancer cell injection, and (ii) a late-stage from 72 h onward [13]. CTX increased cancer cell abundance in the lung at both stages. However, the effect at the late-stage is dependent on a stress-inducible gene *Atf3* in the non-cancer cells in the host (the host-*Atf3*). That is, CTX increased the abundance of cancer cells at the late-stage only in the wild type (WT) mice but not the knockout (KO) mice deficient in *Atf3*. The effect of CTX at the early-stage, in contrast, is not affected by the host genotype.

For the phenotype at the late-stage, we demonstrated that macrophage plays an important role [13]. The CTX-treated WT macrophages were pro-cancer; they protected cancer cells from death and had up-regulated expression of M2 genes, genes expressed in alternatively activated macrophages and in general associated with pro-cancer function. These macrophages allowed the effect of CTX at early-stage (that is, increased cancer cell abundance) to continue and expand. The net result was a higher cancer burden in CTX-treated than control-treated WT mice. In contrast, the CTX-treated KO macrophages were anti-cancer; they were highly cytotoxic to cancer cells and had up-regulated expression of antimicrobial genes. These macrophages erased the effect of CTX at early-stage, resulting in no difference between CTX- and control-treated KO mice. Therefore, macrophage has a dichotomous role, either pro-cancer or anti-cancer, depending on its status of *Atf3*. This dichotomy provides a mechanistic explanation for the phenotype at 72 h (and beyond) that CTX exacerbates lung colonization in a host-*Atf3* dependent manner.

The current study focused on the early-stage phenomenon and addressed how CTX modifies the lung microenvironment to increase the number of cancer cells arrested in the pulmonary endothelium shortly after their arrival. Below, we present evidence supporting the hypothesis that increased vascular adhesiveness is a key mechanism.

## 2. Results

### 2.1. CTX Modified the Lung Endothelium to Allow More Cancer Cells Adhesion—In a Manner Dependent on Host-Derived Serum Factors

As described above, CTX modifies the lung environment, resulting in higher abundance of breast cancer cells in the lung at 3 h, upon their arrival [13]. Using the same model of pre-treatment coupled with intravenous injection of cancer cells (Figure 1a), we found that this phenomenon is also true for paclitaxel (PTX), another frontline chemotherapeutic agent. As shown in Figure 1b (right panel), mice pre-treated with PTX, when compared to vehicle-treated counterpart, had a higher number of cancer burden in the lung at 3 h after intravenous injection of turbo green fluorescent protein (tGFP) labeled MVT-1 breast cancer cells (tGFP-MVT1). For comparison, data from CTX were included on the left, which recapitulated the previous results [13]. In addition, we carried out the 3-h experiment using another breast cancer cell Met1, which differs from MVT-1 in its oncogenic events [14]. Figure 1c shows a similar result. We note that the cell numbers per field of view (FOV) is low, because at this time point (3 h upon cancer cell arrival) most of the cancer cells are eliminated from the lungs by blood flow; yet cell proliferation has not played any appreciable role to increase the number. To enhance the robustness of our data analysis, we implemented several practices (detailed in Materials and Methods, Image Analysis).

Taken together, the ability of chemotherapy to modify the lung to increase the abundance of cancer cells upon their arrival is not restricted to a single chemotherapeutic agent or cancer cell line. Since the half-life in mice is 3–4 h for CTX [15] and 30 h for PTX [16], the majority of the drugs will be excreted by day 4. Therefore, the effect of CTX and PTX is due to their effect on the lung tissue microenvironment, which in turn affects cancer cells.

At 3 h post-injection, most cancer cells have not extravasated (exited the blood vessel to enter the lung parenchyma) [17]. Thus, the cancer cells detected in our assay must be cells with stable intravascular arrest that can resist the blood flow. Both mechanical entrapment and adhesion of cancer cells to the vascular wall have been reported to play a role in intravascular arrest [18,19,20,21,22]. However, discrepancy remains and the molecular details are not well-established (reviews, [23,24,25]). We suspected that entrapment is not the major reason for CTX to increase cancer burden at 3 h in our model, because a single dose of CTX—at four days prior—is unlikely to change the biomechanics of the blood vessel to entrap more cancer cells. Therefore, we hypothesized that the increase in cancer cell number by CTX in our model is due to the change in the vascular wall, making it more adhesive for cancer cells to bind.

As a first step testing this hypothesis, we established an in vitro assay to examine the effect of CTX on the adhesiveness of endothelial cells (ECs). We will refer to this as the “EC adhesiveness assay”. Figure 2a shows the schematic using mouse lung ECs (mLECs). Briefly, mLECs were isolated from neonatal mice to ~90% purity (Figure 2b) using a protocol modified from [26,27] and grown to monolayer, then treated with 4-hydroperoxy-cyclophosphamide (4-OOH) (an active derivative of the prodrug CTX) for 6 h (for the rationale of this treatment period, see Materials and Methods), followed by washing and culturing for 2 days before adding cancer cells. This mimics the in vivo pre-treatment design and avoids the exposure of cancer cells to active CTX. tGFP-MVT1 breast cancer cells were co-cultured with mLECs for 30 min before washing to remove the non-adherent cells, and the remaining cancer cells were counted in a blind fashion (as detailed in Materials and Methods). As shown in Figure 2c, 4-OOH had no effect, indicating that active CTX—by itself—is not sufficient to increase the EC adhesiveness.

One explanation for the negative result in Figure 2c is that the in vitro condition lacks host-derived factors from mice. Therefore, we modified the protocol by injecting CTX into the mice, collecting the serum at 16 h later (Figure 3a), and using the serum (referred to as CTX-serum) in the in vitro “EC adhesiveness assay” (as in Figure 2a). This CTX-serum contains both active metabolites of CTX (from the liver) and any host-derived soluble factors (see Materials and Methods for the rationale of this 16-h design). Figure 3b shows that the CTX-serum increased the adhesiveness of mLECs, compared to the control PBS-serum. Interestingly, the PBS-serum increased mLECs adhesiveness—compared to no serum. Thus, some factors in normal mouse serum can activate mLECs to increase their adhesiveness. We will refer to those factors as the basal-factors. The further increase of mLEC adhesiveness by CTX-serum is due to host factors induced by CTX (referred to as CTX-induced factors), which can be the same as the basal-factors but at higher concentrations, or different (new) factors, or both. We repeated the experiment using human umbilical vein endothelial cells (HUVEC) and found similar results (Appendix A), indicating that the ability of CTX-serum to increase EC adhesiveness is not limited to mLECs.

### 2.2. Matrix Metalloproteinase-2 (MMP-2) Was a Functionally Important Serum Factor for CTX to Increase the Vascular Adhesiveness

To identify the factors in the CTX-serum responsible for increasing EC adhesiveness, we compared CTX- and PBS-serum (pooled from 10 mice each) by protein arrays and found two factors up-regulated by CTX: MMP-2 and osteopontin (OPN, also called SPP1) (Figure 4a and Appendix A). To narrow down the factors relevant to the ability of CTX to increase EC adhesiveness, we took advantage of the observation that CTX increased the abundance of cancer cells in both WT and *Atf3* KO lungs at 3 h after cancer cell injection ([13] and Introduction). Protein array analysis of sera derived from the *Atf3* KO mice showed that MMP-2 and OPN are also up-regulated by CTX (Appendix A). Importantly, they are the only common factors that are up-regulated by CTX in both WT and KO sera (based on the protein arrays used in this study), suggesting them as potential mediators for CTX to increase EC adhesiveness.

We decided to focus on MMP-2 because of the commercial availability of its inhibitors, one of which is MMP-2/9 inhibitor III, a selective inhibitor for MMP-2 and MMP-9 [28]. For convenience, we will refer to it as MMPi in the rest of the report. Because MMP-9 was at a low concentration and was not up-regulated by CTX (Figure 4a, dotted boxes), the effect of MMPi on MMP-9 is not relevant in the context of our experiments. We injected MMPi into the mice after CTX pre-treatment as specified in the Materials and Methods. Figure 4b shows that MMPi almost completely abolished the effect of CTX on cancer cell burden in vivo at 3 h post-cancer injection. Thus, MMP-2, which is up-regulated in mouse serum by CTX, is functionally important for CTX to increase intravascular arrest of cancer cells.

### 2.3. CTX Increased Vascular Permeability in the Lung

One way for CTX to increase vascular adhesiveness is to loosen the EC tight junctions and expose the basement membrane (BM). This is because BM, which is primarily composed of extracellular matrix proteins (ECMs), provides sites for cancer cells to bind and form tight adhesion [29,30]. We will refer to this as the “BM exposure” hypothesis. Since loosening the EC junction would increase blood vessel permeability, we tested this hypothesis by an Evans blue assay. We injected the blue dye intravenously at 4 days after CTX (or vehicle) treatment; 30 min later, we examined the amount of dye that had infiltrated into the lung parenchyma as an indicator for vascular permeability. As shown in Figure 5, CTX increased the intensity of Evans blue in the lung parenchyma. Previously, Liu et al. showed that ischemia increased vascular permeability in an MMP2-dependent manner [31]. Considering the functional importance of MMP-2 in our model (for CTX to increase cancer cell number at 3 h, Figure 4b), we expected that MMP-2 plays a role in the ability of CTX to increase vascular permeability. However, MMPi had no effect (Figure 5). This unexpected result can be reconciled by the differences in our model from that of Liu et al.: lung versus brain ECs (specifically ECs in the blood–brain barrier); CTX versus ischemia; in vivo versus in vitro test of MMPi. This negative result prompted us to ask, “What accounts for the functional importance of MMP-2 in our model”?

### 2.4. MMP-2 Remodels BM and Two ECM Protein Domains Were Required for CTX to Increase Intravascular Cancer Cell Arrest

Data above showed that MMP-2 is required for CTX to increase cancer cell number at 3 h (Figure 4b) but not for CTX to increase vascular permeability (Figure 5). This means that MMP-2 is required at steps after vascular permeability. We proposed that MMP-2 is required to remodel the BM. That is, exposing the BM—by loosening the EC tight junctions—is not sufficient. For CTX to increase cancer cell burden at 3 h, it needs to remodel the BM, making the relevant protein domains accessible for cancer cells to bind. This idea was prompted by the observation that ECM proteins such as laminin and collagen contain cryptic domains that become accessible after proteolytic cleavage [32,33]. To test this remodeling idea, we set up an in vitro binding assay using ECM (matrigel) to mimic the BM and examined the ability of cancer cells to bind. As shown in Figure 6, ECM pre-treated with CTX-serum was more adhesive for cancer cells to bind than the counterpart pre-treated with PBS-serum, indicating that some factors in the CTX-serum modified ECM, making it more adhesive for cancer cells. Importantly, MMPi abolished the effect of CTX-serum, supporting the idea that MMP-2, a CTX-increased factor in the serum, remodels ECM to increase its adhesiveness to cancer cells. This result is consistent with the literature that laminin, a major ECM protein in the BM [34], contains cryptic domains that become accessible after cleavage by MMP-2 [32].

To investigate the protein domains in the BM (a host microenvironment) for cancer cells to bind, we tested two domains: the RGD domain, which is known to interact with the integrin α/β dimers on cancer cells [35,36], and the YIGSR domain, which is known to interact with the laminin receptor (LamR) on cancer cells [37]. Figure 7a shows the sequence of cyclo(RGD) and YIGSR, two peptides that have been shown to block the interaction between ECM and cancer cells [38,39], presumably by acting as competitors. We incubated cancer cells with these peptides before injection and found that they almost completely abolished the CTX effect (Figure 7b). Interestingly, neither peptide alone had any effect (Figure 7c), indicating that cancer cells can latch onto the BM by binding to either domain. Thus, blocking both of them is required to inhibit the CTX effect.

### 2.5. Integrin β1 on Cancer Cells Played an Important Role for Cancer Cells to Interact with the Host Vascular Walls

We next investigated the cell surface proteins that cancer cells could use to interact with the endothelial BM in the lung. As described in Section 2.4, the main binding partner for YIGSR is LamR and for RGD domain is the integrin family of α/β dimers. We tested the potential involvement of these proteins by first profiling the expression of their corresponding genes in the MVT-1 breast cancer cells. Using reverse transcription coupled with quantitative polymerase chain reaction (RT-qPCR), we found that integrin β1 is highly expressed and is the most abundant beta subunit, whereas, α3 and α6 are the most abundant alpha subunits (Figure 8a). This pattern is consistent with the reports [29,40] that integrin dimers α3β1 and α6β1 play an important role in the ability of cancer cells to interact with laminin, a major protein in the BM [34]. The level of LamR mRNA is about 4–5 fold higher than that of integrin β1, consistent with its being a highly expressed gene [41,42]. To test the functional importance of these potential binding partners on cancer cells, we opted to knockdown integrin β1, rather than LamR, for two reasons. (i) It is difficult to efficiently knockdown a highly expressed gene. (ii) The LamR protein precursor is encoded by the ribosomal protein p40 gene [41]. Considering the functional importance of ribosomes, knocking down the p40 gene may interfere with cell viability. To estimate the efficiency of integrin β1 knockdown, we analyzed the cells by flow cytometry and found a reduction from ~85% cells expressing high level of β1 on the cell surface to ~36% (Figure 8b). Functionally, β1 knockdown greatly reduced the ability of CTX to increase cancer cell abundance at 3 h post-injection (Figure 8c). This result is consistent with the literature that integrin β1 plays an important role for cancer cells to interact with laminin [43,44]. An important implication of the knockdown result is that exposing and remodeling BM are not enough. For CTX to manifest its effect, it requires cancer cells to express the proper integrin dimers to take advantage of this host microenvironment change.

The efficacy of integrin β1 knockdown to dampen the CTX effect on cancer burden may appear inconsistent with the results in Figure 7, where blocking the binding partner for integrin β1—the RGD domain—alone was unable to inhibit the CTX effect. Rather, blocking both the RGD and YIGSR domains is required. One potential explanation is that knocking down integrin β1 in cancer cells adversely affects LamR (abundance, localization, or function), rendering it unable to interact with its partner in the host (the YIGSR domain). Therefore, knocking down integrin β1 in cancer cells is sufficient to inhibit cancer cells from forming tight adhesion with the vascular wall, whereas blocking both RGD and YIGSR domains—the binding domains in the host—is required. This supposition is supported by the data that β1-containing integrin dimers co-localize with LamR on the cell membrane [45] and may interact with each other (a review [46]). Consequently, it is possible that integrin β1 knockdown may adversely affect the cell surface LamR. Clearly, more investigation is required to test this supposition.

In summary, to integrate our data, we propose the following two-prong mechanism (Figure 9). (1) CTX increases vascular permeability with an accompanying increase in the exposure of endothelial BM. (2) CTX up-regulates the serum level of MMP-2, which remodels BM, presumably making critical protein domains more accessible. The net result is an increased vascular adhesiveness, allowing cancer cells with proper surface proteins (such as integrin dimers with the β1 subunit) to form tight adhesion, at least in part, by engaging the RGD and YISGR domains, two key protein domains in BM.

## 3. Discussion

Using an unbiased, albeit limited, protein array, we found that CTX increases the serum level of MMP-2. This, combined with clues in the literature, enabled us to elucidate a two-prong mechanism by which CTX modifies the vascular microenvironment in the lung to enhance cancer cell arrest (model in Figure 9). The new finding in our report is 3-fold. (1) CTX increases the systemic level of MMP-2 in the serum and this increase is functionally important for CTX to increase intravascular cancer cell arrest. Our data support the notion that MMP-2 remodels the endothelial BM, making important domains accessible for cancer cells to bind. All these are new. (2) CTX increases vascular permeability. Various stress conditions have been shown to increase vascular permeability. Examples include lipopolysaccharide [47], PTX [48], bleomycin [49], diesel exhaust particles [50], and ischemia [31]. Thus, it is not surprising that CTX, a stressor, increases vascular permeability. However, to our knowledge, increasing vascular permeability by CTX specifically is new. (3) We demonstrated the importance of the YIGSR and RGD protein domains in the host ECM for CTX to increase intravascular cancer cell arrest, and the importance of integrin β1 on cancer cells for this process. Although YIGSR, RGD, and integrin β1 are well described in the literature, the integration of these domains into the context of CTX-enhanced intravascular cancer cell arrest is new. Taken together, we presented data supporting the model in Figure 9. Although some components in the model were reported previously, the model—in its totality—is new.

Various cell types upon activation have been shown to secrete MMPs. They include immune cells (particularly macrophages, [51]), fibroblasts [52], and endothelial cells [53]. In our study, it is not clear what cells are the major sources for the increased serum MMP-2. We note that bone marrow derived cells (BMDCs) from PTX-treated mice were shown to secrete more MMP-9 into the conditioned medium than that from control mice [11]. However, our serum array data did not show any increase in MMP-9. This could be due to differences in chemotherapeutic agents (CTX versus PTX), biological samples (serum versus conditioned medium from BMDCs), and the strains of mice (FVB/N versus C57BL/6). Despite the difference, this report and our data taken together indicate that the induction of MMPs in the host by chemotherapeutic agents is not unique to the experimental condition we used. More investigation is required to address whether this phenomenon is widely applicable. Considering the importance of MMPs in cancer progression [54] and the common use of chemotherapy, this is an important question.

Integrin β1 has been shown to play an important role in various steps of cancer metastatic cascade [55,56]. However, its role in CTX-enhanced intravascular cell arrest has not been reported. As described in the result section, knockdown of integrin β1 greatly dampened the ability of CTX to increase cancer cell arrest. Interestingly, the Met1 breast cancer cells also express a high level of integrin β1 (Appendix A), consistent with the result that CTX also increases their intravascular arrest (Figure 1). On the host side, we showed that the RGD (a binding partner of several integrin dimers with β1 subunit) and YISGR domains are both required for CTX-enhanced cancer arrest. Since both of them are present on laminin [57] and laminin is a major protein in the endothelial BM [34], we posit that laminin is a key ECM protein in the BM for the CTX effect we observed. However, we do not rule out the potential involvement of other ECM proteins in our model. More investigation is required to address this issue.

In addition to increasing cancer cell-BM interaction, CTX can increase intravascular cancer cell arrest by modifying the lung microenvironment in other ways. We examined two possibilities: (a) increasing the cell adhesion molecules (CAMs), which are on the luminal side of ECs and are known mechanisms for cancer cells to adhere to the blood vessel [58,59,60], and (b) activating platelets, which are known to aggregate on disseminated cancer cells in the blood stream and help them adhere to the vascular walls [60,61].

(a) For CAMs, we analyzed the lung ECs at 3 h after cancer injection by flow cytometry to quantify the percentage of ECs expressing the candidate CAMs, and the levels of CAMs per EC (indicated by median fluorescent intensity, MFI). As shown in Appendix A, CTX increased the percentage of ECs expressing N-cadherin, VCAM-1, and P-selectin (panel a), and the abundance of N-cadherin and ICAM per cell (panel b). L-selectin, a lymphocyte-selective CAM [62], had almost non-detectable level in ECs, serving as a negative control. We note that the increase is not dramatic, and preliminary experiments using antibodies to block VCAM-1 or P-selectin failed to dampen the ability of CTX to increase cancer cell arrest (Appendix A). It is possible that simultaneous blocking of more than one CAM is required, or CAM-mediated interaction between ECs and cancer cells is not a major contributor to the CTX effect we observed. (b) For platelet, we inhibited platelet activation by clopidogrel (which blocks ADP signaling [63]) or hirudin (which blocks thrombin activity [64]). Appendix A shows that neither of them inhibited the ability of CTX to increase cancer cell arrest to any appreciable extent. Thus, platelet is likely not a mediator for the effect of CTX in our model.

In summary, we investigated the mechanisms by which CTX increases the adhesiveness of vascular wall for cancer cells to bind. Our data support a two–prong model involving vascular permeability and BM remodeling (Figure 9). These mechanistic insights provide potential future applications to reduce the detrimental effect of CTX. The targets can include MMP-2, integrin β1, or LamR. All these proteins have been the targets of investigation in clinical trials, with varying degrees of efficacy and side-effects [65,66,67]. However, those investigations were in the context of shrinking primary tumors or slowing metastasis, not as a companion treatment to reduce the deleterious effect of chemotherapy. Since chemotherapy is widely used in cancer treatment, reducing its detrimental effect will have significant impact.

## 4. Materials and Methods

### 4.1. Animal Studies

Age- and gender-matched (6–10 week) WT or *Atf3* KO mice in FVB/N background were used for all experiments. KO mice (*Atf3*^−/−^) were generated as described previously [68]) and maintained as separate colonies from the WT mice (originally purchased from Taconic Biosciences). Note that all data in this report were derived from the WT mice, except those in Appendix A (from the *Atf3* KO mice). To avoid potential variations due to genetic drift, every 12–18 months KO mice were backcrossed with their respective WT mice to obtain heterozygotes for two generations before making the appropriate homozygotes. Mice were monitored daily and excluded if they showed any overt signs of stress, illness or fighting. Euthanasia was performed if mice displayed labored breathing, deteriorating body conditions, or inability to ambulate. Mice were euthanized via CO_2_ asphyxiation and cervical dislocation. The rate of mortality for experimental mice was <1%, with no obvious pattern of adverse response to treatment or cancer cell injection. All mice were maintained in climate-controlled, ventilated, and barrier-housing facilities at the Ohio State University, accredited by the American Association for Accreditation for Laboratory Animal Care (AAALAC). Each control and experimental group was housed in their own cages, and all cages under experiment were located on the same rack. JDM and SS were aware of the group allocation during the experiments. For the pre-treatment lung colonization model, mice were intraperitoneally (i.p.) injected with vehicle (2% DMSO in PBS) versus cyclophosphamide (CTX) (Cayman Pharmaceuticals, Ann Arbor, MI, USA) at 150 mg/kg body weight, or vehicle (cremophor EL:ethanol:PBS, 1:1:2) versus paclitaxel (PTX) (Sigma, St. Louis, MO, USA) at 20 mg/kg. Injections of chemotherapeutic drugs were performed in the afternoon between 3–5 PM. Four days later, turbo green fluorescent protein (tGFP)-labeled MVT-1 (tGFP-MVT1) breast cancer cells (10^6^ cells in 100 μL PBS) or tGFP-labeled Met1 (tGFP-Met1) breast cancer cells (2 × 10^6^ cells in 100 μL PBS) were intravenously (i.v.) injected into the tail vein. When indicated, 4 mg/kg MMP-2/9 inhibitor III (MMPi) (Millipore Sigma, Burlington, MA, USA) was injected subcutaneously at 24 and 72 h post-CTX treatment. Lungs were collected 3 h after cancer cell injection for analysis. For inhibitory peptide experiments, 2 mg of YIGSR (Abcam, Waltham, MA, USA) and/or 1 mg of cyclo(RGD) (Bachem, Torrance, CA, USA) were mixed with 10^7^ tGFP-MVT1 cells for 20 min at room temperature. Mice were then injected with 10^6^ each of the peptide-incubated cells, or control cells.

### 4.2. Cell Culture, Treatment, and siRNA Knockdown

tGFP-MVT1 and tGFP-Met1 cells, generated previously [14], were cultured in Dulbecco’s modified Eagle’s medium (DMEM) (Gibco, Waltham, MA, USA) supplemented with 10% fetal bovine serum (FBS), 1% penicillin/streptomycin (P/S) (Gibco), and 3.5 μg/mL puromycin (Sigma). These cell lines express two transgenes (tGFP and puromycin resistance genes), separated by internal ribosome entry site. For CTX treatment in vitro, cells were cultured in a 12-well plate in 1 mL of growth medium containing 10 μM of 4-hydroperoxy cyclophosphamide (4-OOH) (Cayman Chemical, Ann Arbor, MI, USA) or vehicle (0.02% DMSO) for the indicated time before analysis. For the generation of integrin β1 (*Itgb1*) knockdown cells, tGFP-MVT1 cells were grown to 60% confluency in 6-well plates and transfected with 10 μL of 5 μM ON-TARGETplus Mouse *Itgb1* siRNA (Dharmacon, Lafayette, CO, USA) using 10 μL of DharmaFECT reagent (Dharmacon) in 1 mL of Opti-MEM (ThermoFisher, Waltham, MA, USA). Control cells were transfected with ON-TARGETplus non-targeting siRNAs. Transfection media was removed after 24 h, and cells were grown in complete media for another 48 h before use.

### 4.3. Cell Isolation from Mouse Tissues

Mouse lung endothelial cells (mLECs) were isolated via a procedure adapted from two protocols [26,69]. Briefly, lungs from 7–8 day old FVB/N neonatal mice were aseptically harvested and placed into Miltenyi C-tubes (Miltenyi Biotec, Auburn, CA, USA) with 4 mL of a collagenase (Sigma) and dispase (Sigma) solution at 1 mg/mL each and then minced into 2–3 mm pieces. This mixture was incubated at 37 °C with constant agitation in a slowly oscillating water bath for 30 min, whereupon DNase I (Sigma) was added to a final concentration of 25 units/mL, followed by 30 min of incubation. The tissue was then pulverized using a gentleMACS Dissociator (Miltenyi Biotec). The resulting slurry was passed through a 100 μm filter. Cells from three neonatal lungs were pooled and pelleted at 500× *g* at 4 °C and then resuspended in a 1X ammonium chloride red blood cell lysis buffer (15.5 mM NH4Cl, 1 mM KHCO3, 0.01 mM EDTA). After 10 min at room temperature, 10 volumes of water were added and the cells were pelleted again at 500× *g* and resuspended in 90 μL of MACS buffer (0.5% bovine serum albumin and 2 mM EDTA in PBS) to generate single cell suspension (in general, ~3 × 10^6^ cells/neonatal lung) and mixed with 10 μL of CD31 magnetic beads, washed, and eluted in 1 mL of PBS as recommended by the manufacturer (Miltenyi Biotec). About 5 × 10^5^ cells were recovered per neonatal lung, and ~1–3 × 10^6^ bead-enriched cells were cultured on 2% gelatin-coated 10-cm plates with EndoGRO LS medium (Sigma) containing 1% P/S for three days to reach confluence, at which time the purity was in general ~90% CD31^+^ by flow cytometry. mLECs were used for only three passages following isolation.

### 4.4. Generation of CTX-Serum and PBS-Serum

FVB/N mice were injected with PBS or CTX, and 16 h later, blood (500 μL per mouse) was collected by cardiac puncture and left at room temperature for 30 min to clot, followed by centrifugation at 500× *g* for 10 min at 4 °C. The resulting supernatant (~200 μL) was collected as serum, and stored at −80 °C. They are referred to as PBS- or CTX-serum. To reduce biological variation, sera from 3–4 mice were pooled for each experiment. The 16 h time point was chosen to allow time for CTX to be metabolized by the liver, and for secreted molecules (if any) to accumulate in circulation in response to CTX. The supposition we aimed to test was that some important host serum factors are present at this time point, which set things in motion to change the lung microenvironment, contributing to increased intravascular cancer cell adhesion.

### 4.5. In Vitro EC-Cancer Cell Adhesion Assay

mLECs or human umbilical vein endothelial cells (HUVECs) were grown in 12-well plates until confluent. Once the formation of monolayers was confirmed by light microscopy, cells were treated with medium containing 4-OOH (10 μM) or vehicle (0.02% DMSO), or PBS- or CTX-serum diluted 4-fold in medium for 6 h. Wells were then washed three times with PBS, and cultured with complete media for two or four days (as specified in the figure). At that time, tGFP-MVT1 cells were added to the wells. Thirty minutes later, non-adherent cells were washed away with PBS, and the remaining cells imaged using an inverted fluorescent microscope, followed by image analyses described in 4.7. The rationale of treating ECs for 6 h is partly based on the pharmacokinetics of CTX in mice (half-life 3–4 h) and partly arbitrary (~2 half-lives). This mimics the in vivo situation that the host cells are exposed to the drug only “transiently,” not for a pro-longed period. The concentrations of 4-OOH and serum were determined by pilot experiments to obtain the highest dose without cytotoxicity.

### 4.6. Protein Array

Serum from WT and *Atf3* KO mice was collected as described above in Section 4.4, and diluted 5-fold in blocking buffer provided in the kit (RayBiotech, Peachtree Corners, GA, USA). One ml of the diluted sample was placed onto blocked Mouse Cytokine Arrays C3 and C4 and processed according to the manufacturer’s instructions (RayBiotech). Array membranes were imaged using a Sapphire Biomolecular Imager (Azure Biosystems, Dublin, CA, USA) and densitometry measurements were made using AsureSpot Analysis Software. To calculate changes in protein levels between groups, intensity values were first background-corrected using negative control spots and then normalized against the average intensity of the 6 positive control spots.

### 4.7. Image Analysis

For image analyses of in vivo experiments, survey the lung sections to capture representative images, and collect 6–10 images (at 100× magnification) per section. Images from all groups of mice were coded, pooled, and randomized using the “Random Names” tool by Jason Faulkner. The purpose of randomization is to eliminate any clues the investigators may have while analyzing the images, if they are clustered together based on their groups. In general, we analyze 100–200 images in one setting. To exclude background noise, the investigators (JDM and SS) followed the same criteria: designate discrete, bright green fluorescent signals with TO-PRO-3 nuclear stain as cancer cells, and exclude diffuse and faint fluorescent areas that lack nuclear stain as background noise. These backgrounds were presumably due to auto-fluorescence of bronchioles and/or blood vessels, and were more evident in some set of experiments than others. As indicated in the figure legends, multiple mice were used per group and the experiments were repeated. These should help even out potential inconsistency in judgement calls to exclude backgrounds.

For image analysis of in vitro cell adhesion assays (on ECs or matrigel), take non-overlapping images to cover the entire well for 96-well plates, or survey the well and take 5 representative images for 12-well plates. Employ the same principles as above to avoid bias and reduce inconsistency: code the images, pool images from different groups, randomize them before analysis, use multiple samples per group, and repeat the experiments. The threshold was set to remove the background noise that was faint in intensity or small in size (debris).

Analyses for both in vivo and in vitro assays were done using Zeiss Zen Lite software or FIJI where appropriate.

### 4.8. RNA Isolation and RT-qPCR

RNA was isolated from the indicated cells using TRIzol (ThermoFisher) according to the manufacturer’s instructions. Concentration and purity were determined by NanoDrop 2000 (ThermoFisher). RT-qPCR was performed as previously described [70] using glyceraldehyde 3-phosphate dehydrogenase (*Gapdh*) as an internal control. All primers are listed in Appendix A.

### 4.9. Immunofluorescence Analysis

Staining was performed on formalin-fixed, paraffin-embedded tissue sections (10 μm thickness) as previously described [71]. Briefly, slides were rehydrated, blocked with 0.3% hydrogen peroxide in methanol, washed in Tris-buffered saline +0.1% Tween 20 (TBST) and blocked in 5% normal horse serum for an hour at room temperature. Primary antibodies were applied at the indicated concentrations (Appendix A) overnight at 4 °C. Slides were then incubated with ImmPress secondary antibodies (Vector) for an hour, followed by the application of a tyramide signal amplification system (ThermoFisher) and TO-PRO-3 (nuclear stain) for 15 min, both at room temperature. Slides were mounted in Vectashield (Vector) and the images were captured on a Leica TCS SL confocal microscope (Leica, Wetzlar, Germany), an Andor Spinning Disk confocal microscope (Oxford Instruments, Abingdon, UK), or a Zeiss LSM800 confocal super-resolution microscope (Carl Zeiss Microscopy, White Plains, NY, USA) (depending on availability).

### 4.10. Evans Blue Vascular Permeability Assay

Four days after injection of PBS or CTX, mice were i.v. injected with 150 μL of 1% Evans blue dye (Sigma) solution in PBS. After 30 min, mice were euthanized and perfused with PBS until the perfusate became clear, to remove the dye within the blood vessels as thoroughly as possible. Lungs were minced and incubated with 1 mL of formamide for 48 h in a 60 °C water bath. Samples were then centrifuged at 2000× *g* for 30 min, and the supernatant measured for absorbance at 620 nm with normalization against a blank control (formamide only). Evans blue concentrations were ascertained based on a standard curve and used to calculate the total amount of dye extracted from each lung parenchyma. The lung pellet was dried in a SpeedVac overnight, and the weight of the dried lungs was used as a denominator to calculate the nanogram of Evans blue per gram of dried lung. When indicated, MMPi was injected as described in Section 4.1.

### 4.11. Analytical Flow Cytometry and Fluorescence Activated Cell Sorting (FACS)

Single cell suspensions from lungs were prepared as described in Section 4.3. Afterwards, 2 × 10^6^ cells were incubated with CD16/32 Fc blocking antibodies for 10 min at 4 °C in 100 μL of flow buffer (5% FBS in PBS). Cells were then pelleted in a refrigerated centrifuge at 500× *g* for 5 min, resuspended in flow buffer, and stained with the indicated antibodies (Appendix A) for 15 min at 4 °C. Antibodies were washed away, and the cells were fixed in 2% formalin (in PBS) for at least 30 min, pelleted and resuspended in flow buffer before analysis using BD LSR II or Fortessa flow cytometers (Becton, Dickenson and Co., Franklin Lakes, NJ, USA). For FACS, cells were run on a BD FACSAria III or a BD FACSAria Fusion on a purity setting. Unstained and single-stained cells were used as gating controls and for compensation.

### 4.12. In Vitro Matrigel (ECM)-Cancer Cell Adhesion Assay

Matrigel (Corning, Tewksbury, MA, USA) was diluted in serum- and antibiotic-free DMEM to 1 mg/mL and used to coat the plate. A total of 100 μL of the diluted matrigel was deposited into each well of a 96-well plate and then incubated at 37 °C for 2 h. Afterwards, excess liquid was removed and 100 μL of DMEM or serum from mice (PBS- or CTX-serum diluted 1:100 in DMEM, a concentration determined by pilot experiments) were added to wells, with or without 50 μM MMP-2/9 inhibitor III (Millipore Sigma), and incubated for 6 h. Media/serum was removed, and 20,000 tGFP-MVT1 cell s were added in 200 μL of complete DMEM, and incubated for 30 min. Cells were then removed and plates were washed 3 times with PBS and the remaining cells imaged on a Leica microscope with a 2.5× objective. Cells were counted using FIJI’s “Analyze Particles” program with the minimum size of signals/cells set at 50 μm^2^.

### 4.13. Statistics

Sample size was calculated based on the standard deviation in previously published results [5] or from pilot experiments, with the expectation to detect a 20% difference (2-sided) with 80% power and 95% confidence. Data were analyzed using GraphPad Prism 6.0 (for the generation of figures) and SigmaPlot 13.0 (for statistics). All data represent mean ± standard error (SE). A *p* value of less than 0.05 was considered statistically significant. Student’s *t* test, one-way ANOVA and two-way ANOVA were used as indicated. A post hoc Holm–Šídák correction to counteract family-wise error rate was used for all ANOVA tests and normality was assessed using the Shapiro–Wilk test. Grubb’s test was used to detect outliers and none was found for all data reported here.

## 5. Conclusions

Data in this report provide a two-prong mechanism for CTX to modify the lung microenvironment: (a) an increase in vascular permeability, and (b) an MMP2-mediated remodeling of endothelial basement membrane (BM). The net result is an increase in vascular adhesiveness, allowing cancer cells—shortly after their arrival at the lung—to form tight adhesion to the BM. The critical protein domains for this tight adhesion are the RGD and YIGSR domains in the BM, which are known binding partners for integrin dimers and laminin receptor on cancer cells. Vascular adhesion is the first step for the circulating cancer cells to colonize the secondary site. Since tissue colonization is a rate-limiting step in metastasis (the major cause of cancer death), cancer cell adhesion to the vascular wall is an important step. Understanding how chemotherapy may change the non-cancer cells in the organism to enhance this critical step in metastasis may provide strategies to improve the efficacy of chemotherapy.

## Figures and Tables

**Figure 1 ijms-22-10280-f001:**
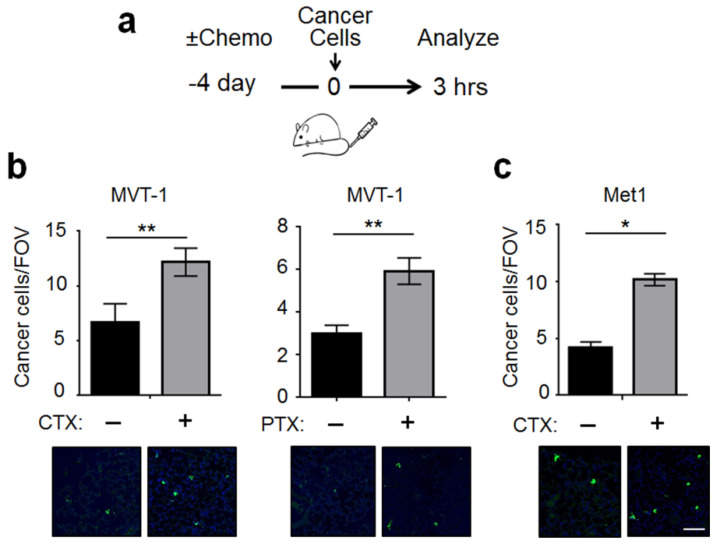
Chemotherapy pre-treatment modified the lung microenvironment to increase its adhesiveness to cancer cells. (**a**) A schematic of the lung colonization model with chemotherapeutic agent (+) or vehicle (−) treatment on day 4 before tail vein injection of the indicated breast cancer cells. (**b**) *Left*: Quantification of lung cancer cells per field of view (FOV) in mice pre-treated with CTX or vehicle. *Right*: Same as the left panel except PTX and its vehicle were used. Bottom: Representative images (showing a portion of the FOV). (**c**) Same as the left panel in (**b**) except the tGFP-Met1 breast cancer cells were used (*n* = 6–9 from 2–3 independent experiments). Scale bar: 50 μm. Bars indicate mean ± SEM; Student’s *t* test; * *p <* 0.05; ** *p <* 0.01.

**Figure 2 ijms-22-10280-f002:**
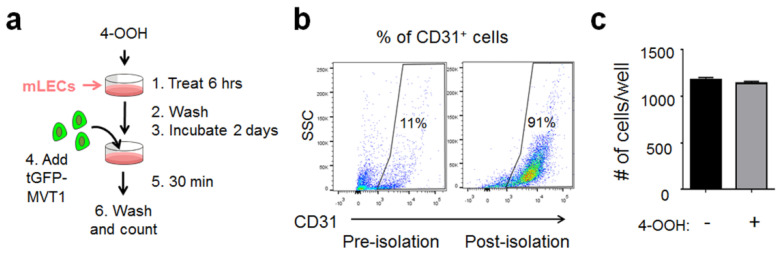
Active CTX was unable to modify the mLECs to allow more cancer cell adhesion in vitro. (**a**) A schematic of the in vitro cancer cell-mLEC adhesion assay. (**b**) Shown is a representative mLEC purity determined by CD31 positivity via flow cytometry. (**c**) Quantification of tGFP-MVT1 cells per well (in 96-well plates) adhered to mLECs pre-treated with 4-hydroperoxy-cyclophosphamide (4-OOH) (+) or vehicle (−) (*n* = 6 from 2 independent experiments). Bars indicate mean ± SEM; Student’s *t* test.

**Figure 3 ijms-22-10280-f003:**
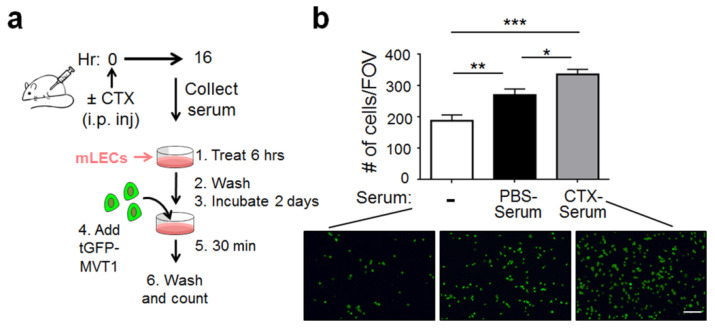
CTX-serum increased the mLEC adhesiveness in vitro. (**a**) A schematic. (**b**) *Top*: Quantification of tGFP-MVT1 cells per FOV adhered to mLECs pre-treated with CTX-serum, PBS-serum, or medium only (−) (*n* = 13–14 from 3 independent experiments). Total 360 images from 3 groups were pooled, randomized and scored in blind. *Bottom*: Representative images. Scale bar, 100 μm. Bars indicate mean ± SEM; one-way ANOVA with post hoc Holm–Šídák correction; * *p <* 0.05; ** *p <* 0.01; *** *p <* 0.001.

**Figure 4 ijms-22-10280-f004:**
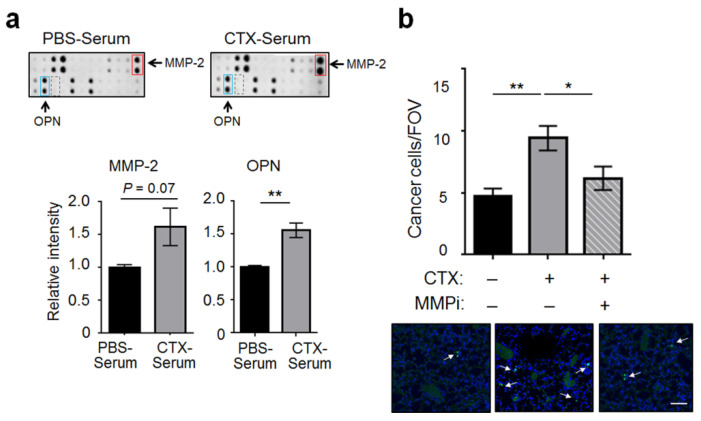
CTX increased the serum level of MMP-2, which was functionally important for CTX to increase the adhesiveness of vascular walls. (**a**) *Top*: A portion of the protein array is shown for MMP-2 (red box), OPN (blue box), and MMP-9 (dashed box). *Bottom*: Signals of MMP-2 and OPN were normalized as detailed in Materials and Methods (subtract the background and normalized against the positive controls). The normalized signals from the PBS-serum blot were arbitrarily defined as 1. (**b**) Mice were pre-treated with CTX (+) or vehicle (−), followed by injection of MMP-2/9 inhibitor III (MMPi) (+) or vehicle (−) 24 and 72 h after CTX treatment. Cancer cell number per FOV at 3 h post-injection were analyzed (*n* = 6–9 from 2 independent experiments). *Bottom*: Representative images. Arrows indicate cancer cells, which are discrete and bright, in contrast to the background noise of faint and diffuse nature (presumably due to the auto-fluorescence of bronchioles and/or blood vessels). Scale bar, 50 μm. Bars indicate mean ± SEM; one-way ANOVA with post hoc Holm–Šídák correction; * *p* < 0.05; ** *p* < 0.01.

**Figure 5 ijms-22-10280-f005:**
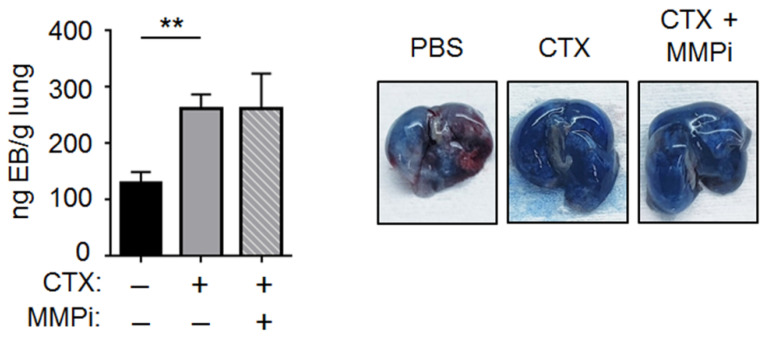
CTX increased vascular permeability in the lung. *Left*: Mice were pre-treated with CTX (+) or vehicle (−), and 4 days later their vascular permeability in the lung was assayed by Evans blue (EB) dye as detailed in Materials and Methods. When indicated, MMPi was injected 24 and 72 h after CTX treatment. After removing intravascular EB by perfusion, EB in the lung parenchyma was extracted by formamide and the nanogram (ng) EB per gram (g) of dried lung was determined as described in Materials and Methods (*n* = 4–8 from 2 independent experiments). *Right*: Representative images. Bars indicate mean ± SEM; one-way ANOVA with post hoc Holm–Šídák correction; ** *p* < 0.01.

**Figure 6 ijms-22-10280-f006:**
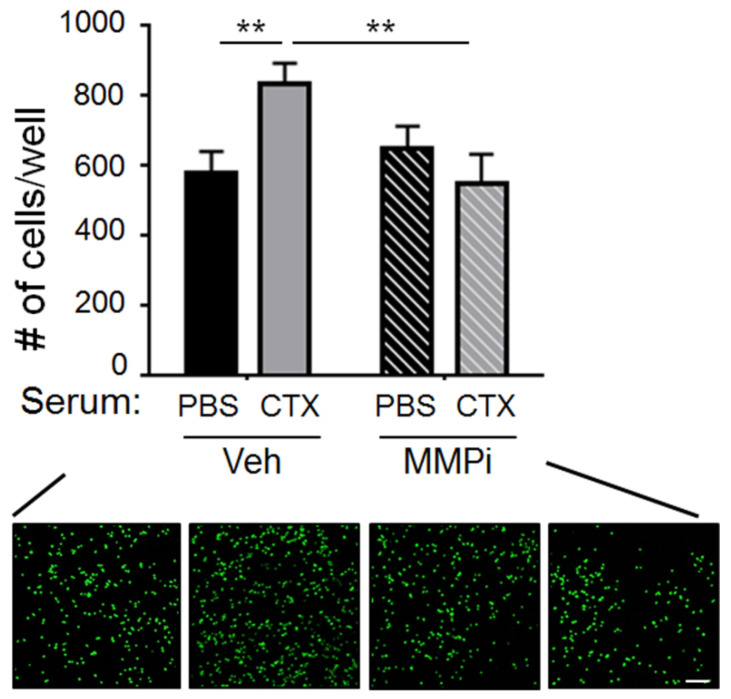
CTX-serum increased the binding of cancer cells to ECM in vitro. Plates coated with matrigel were pre-treated with PBS-serum or CTX-serum in the presence of MMPi or vehicle (Veh) before incubation with tGFP-MVT1 cells to assay their ability to allow cancer cell adhesion. *Top*: Quantification of cells per well after the indicated treatments (*n* = 20 from 4 independent experiments). *Bottom*: Representative images. Scale bar, 100 μm. Bars indicate mean ± SEM; two-way ANOVA with post hoc Holm–Šídák correction; ** *p* < 0.01.

**Figure 7 ijms-22-10280-f007:**
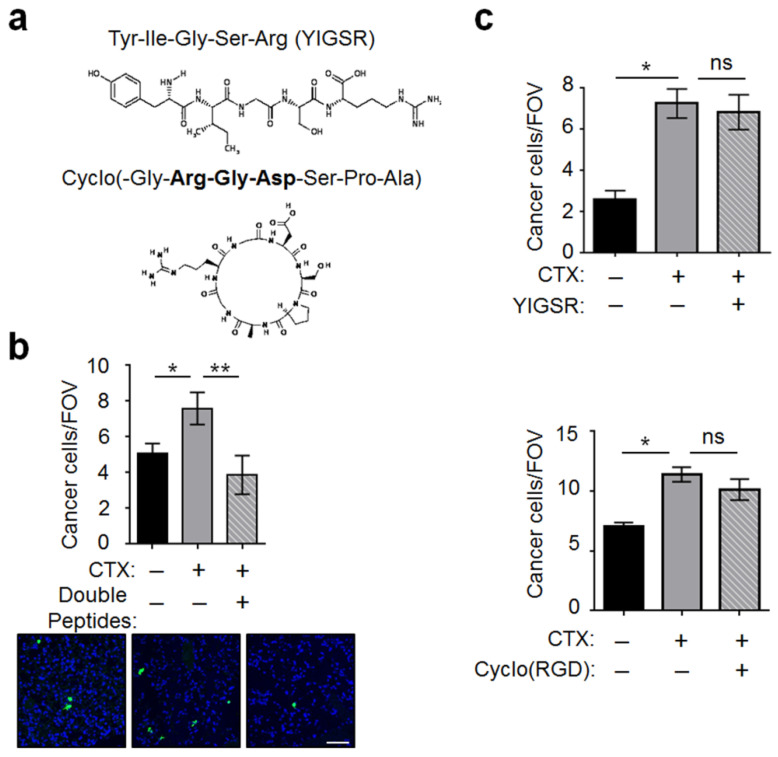
The YIGSR and cyclo(RGD) peptides derived from ECM proteins dampened the ability of CTX to increase vascular adhesiveness. (**a**) Schematics and sequences of the peptides. (**b**) Cancer cells were incubated with the YIGSR and cyclo(RGD) double peptides for 20 min before injection into mice pre-treated with CTX (+) or vehicle (−). Cancer cell number per FOV at 3 h after injection were analyzed (*n* = 8–9 from 2 independent experiments). *Bottom*: Representative images. (**c**) Same as in (**b**) except single peptide was used: YIGSR (top) and cyclo(RGD) (bottom) (*n* = 6–8 from 2 independent experiments). Scale bar, 50 μm. Bars indicate mean ± SEM; one-way ANOVA with post hoc Holm–Šídák correction; * *p* < 0.05; ** *p* < 0.01; ns: not significant.

**Figure 8 ijms-22-10280-f008:**
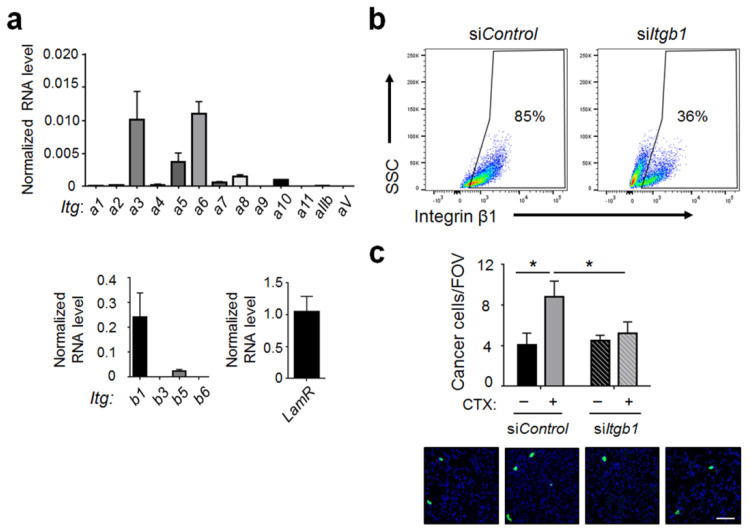
Integrin β1 knockdown in cancer cells reduced the effect of CTX. (**a**) mRNA levels of laminin receptor (*LamR*) and the indicated integrins (*Itg*) in tGFP-MVT1 cells were analyzed by reverse transcriptase coupled with quantitative polymerase chain reaction (RT-qPCR). Signals were normalized against glyceraldehyde 3-phosphate dehydrogenase (*Gapdh*) (from 4 independent experiments). (**b**) tGFP-MVT1 cells with *Itgb1* knockdown (si*Itgb1*) or control knockdown (si*Control*) were analyzed for their integrin β1 level on the cell surface by flow cytometry to estimate the knockdown efficiency. (**c**) tGFP-MVT1 cells with si*Control* or si*Itgb1* knockdown were injected into mice with CTX (+) or vehicle (−) pre-treatment. Lung cancer cells per FOV at 3 h post-injection were analyzed (*n* = 6–8 from 2 independent experiments). *Bottom*: Representative images. Scale bar, 50 μm. Bars indicate mean ± SEM; for panel (**c**), two-way ANOVA with post hoc Holm–Šídák correction; * *p* < 0.05.

**Figure 9 ijms-22-10280-f009:**
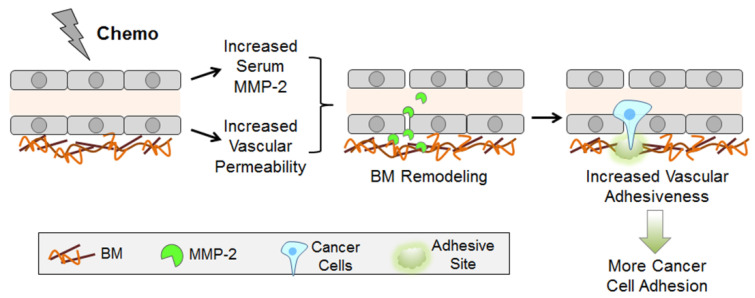
A model. See text for details. BM: basement membrane.

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
