# Peer review of "Chemotherapy-Induced Changes in the Lung Microenvironment: The Role of MMP-2 in Facilitating Intravascular Arrest of Breast Cancer Cells"

_ijms, 2021, doi:10.3390/ijms221910280_

Round 1

Reviewer 1 Report

This study by Justin D. Middleton and colleagues explored cyclophosphamide-induced changes in the lung microenvironment and the role of MMP-2 in facilitating intravascular arrest of breast cancer cells.

Questions and suggestions:

  1. The bar graphs in Figure 1b should accompany with representative images.
  2. Can the authors explain the source of MMP-2 in serum in early stage of the lungs?
  3. It is important to look at immune cells such as macrophages and neutrophils etc. changes after CTX treatment. Have you analyzed immune profiling of the lungs in CTX-treated mice? 
  4. For the in vivo data, authors need to make it clearer regard to two genotypes WT and Atf3 KO mice in every figures. 
  5. Did the author exclude the dead cells from flow cytometry analysis in Figure 8b? Can you please show the FMO sample in this flow cytometry analysis?

Author Response

REVIEWER #1:

Critique 1: The bar graphs in Figure 1b should accompany with representative images.

Response: We added the representative images as requested. Please note that each image only shows a portion of the FOV. Therefore, the number of cells in the image does not match that shown in the Y-axis. The reason for only showing a portion of the FOV is to enlarge the size of each positive signal (for easier inspection). For detailed description of our image analyses, please see the response to Reviewer #2, Critique 1.

Critique 2: Can the authors explain the source of MMP-2 in serum in early stage of the lungs?

Response: As discussed in the manuscript, various cell types can be the source of MMP2, such as immune cells, fibroblasts, and endothelial cells. We agree with the reviewer that it is of interest to identify the cell source. However, this is not a trivial issue to address. (a) CTX is metabolized in the liver and the active metabolites circulate throughout the body to increase the systemic level of MMP2. It is not clear which organ is the best one to harvest for analyses, let along which cell type. (b) The increase in serum level of MMP2 could be due to increased gene expression (higher mRNAs and/or protein levels in the cells), or increased secretion of pro-MMP2, or both. Together, they—organ(s), cell type(s), and biological process(es)—make this a non-trivial problem to solve. A case in point, we analyzed the lung endothelial cells (post-CD31 enrichment) and found no increase in MMP2 mRNA level after CTX treatment (data not shown). Therefore, despite the importance of this question, we believe that it is beyond the scope of our current study and will be best addressed by systematic and careful analyses in the future.

Critique 3: It is important to look at immune cells such as macrophages and neutrophils etc. changes after CTX treatment. Have you analyzed immune profiling of the lungs in CTX-treated mice?

Response: This is an interesting experiment. Although we have not done this exact experiment, we carried out immunophenotyping on day 4 after CTX treatment, before cancer cell injection. The markers we used were CD11b, Ly6C, Ly6G, F4/80, and VEGFR1. This combination allows us to assess various cell populations, including monocytes, neutrophils, and macrophages. Our preliminary results showed a trend of increase in these cells by CTX, with neutrophil showing the most obvious increase—but not yet statistically significant (probably due to the small size of this population). For inspection by the reviewers, we include a representative t-distributed stochastic neighbor embedding (t-SNE) map and bar graphs (see figure in the uploaded letter of response). Clearly, more studies are required to see whether the differences are reproducible and whether they can achieve statistical significance. If so, what is their functional significance? Are they relevant to the intravascular cancer cell adhesion described in our report? These are beyond the scope of this study.

Critique 4: For the in vivo data, authors need to make it clearer regard to two genotypes WT and Atf3 KO mice in every figures.

Response: All data were derived from WT mice, except Figure S3. The genotype of the mice (Atf3 KO) is labeled in Figure S3 and indicated in its legend. In response to this comment, we clarified this point in the M&M section.

Critique 5: Did the author exclude the dead cells from flow cytometry analysis in Figure 8b? Can you please show the FMO sample in this flow cytometry analysis?

Response: We added a figure for gating strategy (Figure S4b), which shows the exclusion of debris and cell aggregates by FSC/SSC, and the fluorescence minus one (FMO) control (no integrin beta1 staining). In routine experiments, we found that the MVT-1 cells harvested from cell culture are consistently >95% viable (see figure in the uploaded letter of response). 

Reviewer 2 Report

This manuscript described the relationship between chemotherapy and the breast cancer microenvironment. The authors are trying to demonstrate that the increased accessibility of critical protein domains in the basement membrane with the higher vascular adhesiveness for cancer cells to adhere. Overall, this is an interesting study demonstrating some interesting results. However, there are several issues with the manuscript in its current form which require attention before it can be recommended for publication.

Here are some comments:

  1. In Figure 1/4/7, Could the authors provide a representative image (in vivo assay) between the control and treatment groups? From the bar chart, the cells/FOV is quite low. How confident do they feel you are counting the cells, not the noise? If they can stain the nuclei as well that would be better.

  1. In the “EC adhesiveness assay” (Figure 2) and the “modified EC adhesiveness assay” (Figure 3). The authors gave the explanation for the negative result in Figure 2c is that the in vitro condition lacks host-derived factors from mice. Therefore, they used the “treated/untreated” mice serum instead of the drugs to perform the modified experiment. I believed this should be the only difference between the experiments. However, from the result, there is a big gap between the number of cells/FOV in the negative control group (~1200 vs. ~200). How would the authors explain this difference (especially since this is an in vitro assay)?

  1. In Figure 3b, the representative pictures have different background fluorescence intensities. To be honest, I could not see the difference if I adjust the bright/contrast of the pictures between the negative and PBS-serum groups. It just looks like some of the cells have a lighter colour (but they are definitely there). I am not sure whether this will affect the quantification when you use Fiji.

  1. In Figure S1, the experiment is the same as Figure 3 but using the HUVEC. Why the quantification of the images using Pixels rather than cell numbers which is not consistent with the rest of the manuscript?

  1. In Figure 4, the authors selected MMP-2 with a p-value of 0.07. I am not saying p>0.05 would not be considered for further investigation. However, when you described in the manuscript, it may be better to use some other statement (e.g approached the borderline of significance?) rather than saying significant difference.

  1. In Figure 7b, the middle bar should be CTX- Peptide+

Author Response

REVIEWER #2:

Critique 1: In Figure 1/4/7, Could the authors provide a representative image (in vivo assay) between the control and treatment groups? From the bar chart, the cells/FOV is quite low. How confident do they feel you are counting the cells, not the noise? If they can stain the nuclei as well that would be better.

Response:

We added representative images to Figures 1 and 4 (in vivo assay), and Figure 7 (in vitro assay).

To address the second half of the critiques regarding the validity of the data, we provide the details of our image analyses below.

In vivo assay for cancer cells in the lung:

  • At 3 hours after cancer cell injection, the cell number is very low. This is because most injected cells are eliminated by the blood flow; yet cell proliferation has not played a significant role to increase the cell number.
  • To increase the validity of our analyses, we implemented the following practices.
  • Survey each lung section and take pictures of the representative areas. Use 100x (a relatively low magnification) to capture more “events” (cells) per field of view (FOV).
  • Use multiple mice per group; take minimally 6 images per mouse; code the images and pool them together; randomize the images (to eliminate any clues the investigators may have if the images are clustered together based on their groups); repeat the experiments. In general, we score 100-200 images in one setting.
  • Follow consistent criteria to exclude background signals. Close inspection of the images reveals some diffused, light green fluorescent areas—in addition to the discrete and bright tGFP-labeled cancer cells. Note that the background is more obvious in some images than others. These diffused areas are likely bronchioles and/or blood vessels due to their auto-fluorescence. They can be easily distinguished from the tGFP cells by their diffused nature, low signal intensity, and lack of TO-PRO-3 nuclear stain. The two investigators who scored the cells (JDM and SS) discussed the issues and used the same criteria.

The pooling and randomization steps prevent potential biases (conscious or unconscious); the multiple samples and repeated experiments help even out the variations due to inconsistent judgement calls in excluding the background. Together, they increased our confidence of the data—even though the cell number was relatively low at the 3-hour time point.

  • Note that each image only shows a portion of the FOV. Therefore, the number of cells in the image does not match that shown in the Y-axis. The reason for only showing a portion of the FOV is to enlarge the size of each positive signal (for easier inspection).

In vitro assay for adhered cancer cells:

  • For PBS- or CTX-serum, pool sera from 3-4 mice per group to reduce biological variations.
  • Take pictures of the non-overlapping images to cover the entire well.
  • Employ the same principles used for the in vivo assay (above): multiple samples per group, pooling images from different groups, randomizing the pooled images, and repeating the experiments.
  • Follow consistent criteria to exclude background noise by setting the threshold to remove faint intensity and debris of small size. Count the cell numbers automatically using FIJI.
  • Only a portion of the well is shown in the representative images.

In response to the critique, we added not only representative images but also more details for image analysis (in M&M and text).

Critique 2: In the “EC adhesiveness assay” (Figure 2) and the “modified EC adhesiveness assay” (Figure 3) … from the result, there is a big gap between the number of cells/FOV in the negative control group (~1200 vs. ~200). How would the authors explain this difference (especially since this is an in vitro assay)?

Response: We thank the reviewer for pointing out this discrepancy. It is a mistake. Figure 2’s Y-axis is “Cell # per well,” but was mis-labeled as “Cell # per FOV.” We corrected the mistake.

Critique 3: In Figure 3b, the representative pictures have different background fluorescence … looks like some of the cells have a lighter colour (but they are definitely there). I am not sure whether this will affect the quantification when you use Fiji.

Response: As indicated in our response to Critique 1, the investigators set threshold in a blind manner before FIJI analysis. The pooling and randomization steps prevent bias, and the multiple samples and repeated experiments help even out variations due to potential inconsistency in judgement calls. All these contribute to the robustness of the data. However, the reviewer’s comment is well-taken; we re-examined the data and replaced the panel with images of similar background.

Critique 4: In Figure S1, the experiment is the same as Figure 3 but using the HUVEC. Why the quantification of the images using Pixels rather than cell numbers which is not consistent with the rest of the manuscript.

Response: That set of data was obtained at different time. In response to the critique, we re-analyzed the data and changed the Y-axis to Cell #/FOV.

Critique 5: In Figure 4, the authors selected MMP-2 with a p-value of 0.07. ... when you described in the manuscript, it may be better to use some other statement … rather than saying significant difference.

Response: We edited the text.

Critique 6: In Figure 7b, the middle bar should be CTX- Peptide +.

Response: We thank the reviewer for pointing out the mistake and corrected it.

Reviewer 3 Report

The objective of the authors was to investigate how pre-treatment with chemotherapeutic drug cyclophosphamide (CTX) increase cancer cell burden in the lung through modulation of the lung microenvironment. The authors have reported CTX modulate lung microenvironment through increased vascular adhesiveness and permeability by regulation of MMP2 levels and through MMP-2 mediated remodeling of the endothelial basement membrane. The data is very interesting but the manuscript could not be considered for publication in the current form for following reasons.

Major concerns:

1) Could authors please explain the rational for selection of 6 hrs treatment window to treat with mLECs with 4-OOH? Did authors choose this treatment period through any pilot studies or followed the recommendations from previous studies? If so please indicate the reference.

2) Did authors by any chance examine whether longer treatment  of mLECs with 4-OOH could alter breast cancer cells adhesion in comparison to 4 hr treatment of mLECs with 4-OOH in the presence or absence of CTX-serum?

3) For Figure 3 and Figure S1, the authors have used serum collected 16hr post treatment with chemotherapeutic agent, CTX. Did authors treated mLECs and HUVEC with CTX-Serum for 4 hours and washed the cells and incubated in regular media for next four days before adding the breast cancer cells?

4) If authors have followed above experimental strategy in comment#3 could authors please clarify the rational for not maintaining the cells in CTX-serum throughout culture period as host-secreted factors are considered as the basis for the inability to recapitulate in vivo findings as mentioned in Figure 2 data. 

5) Did author perform any studies to examine the difference in maintaining cells with and without CTX serum till the study endpoint on adhesion of cancer cells.

6) The authors have stated 16 hr time window was chosen to ensure accumulation of any secreted molecules in circulation in response to CTX. If host derived factors were the reason, the authors should have collected at a later time point such as at 96h to be consistent with their rational on not only the clearance of the chemotherapeutic agent from the system but also to rule out the impact of CTX on signaling cascade? Could authors please provide comment on it.

7) Could authors please include additional details in schematic diagram of Figure 3. It may be redundant of Fig2 but would provide clarity on overview of the experiment procedure followed to the readers. 

8) Could authors please clarify whether chemotherapeutic agents were administered intraperitoneally or through intravenous injections. In materials and methods the authors have mentioned the route of administration as I.P but in Figure 3A and Figure S1 the route of administration is mentioned as I.V. Please look into it and correct it.

Minor concerns:

1) The data shown in Figure 1B is representative of data from 3hr post injection of respective breast cancer cell lines? The authors have used 3hrs as the end point for the studies. So could authors please have the arrow before 3 hrs and remove the arrow after 3 hrs to avoid ambiguity of other timepoints.

Author Response

REVIEWER #3:

Critique 1: Could authors please explain the rational for selection of 6 hrs treatment window to treat with mLECs with 4-OOH? Did authors choose this treatment period through any pilot studies or followed the recommendations from previous studies?

Response: The half-life of CTX in mouse is about 3-4 hours (reference 15 in the manuscript). This means that the concentration of CTX that the host cells encounter is reducing continuously due to CTX elimination (excreted from the mice). For the in vitro assay, we arbitrarily selected 6 hours (~2 half-lives) to mimic the “transient” drug exposure in vivo. Therefore, the experimental design was partly based on the pharmacokinetics of CTX in mice and partly arbitrary (~ 2 half-lives). However, we did carry out a dose curve study (1, 5, 10, 20, 50 mM) and found that 10 mM is the highest dose without causing toxicity. Hence, we used this dose in the study. In response to this critique, we edited the manuscript to clarify these points.

Critique 2: Did authors by any chance examine whether longer treatment of mLECs with 4-OOH could alter breast cancer cells adhesion … ?

Response: No, we treated the cells” transiently” to mimic the in vivo situation. See our response to Critique 1. We note for all in vitro assays (using HUVECs or mLECs), the treatment (by 4-OOH or serum) was 6 hours. The 4 hour label in one figure is a mistake and is corrected in the revision.

Critique 3: For Figure 3 and Figure S1, … Did authors treated mLECs and HUVEC with CTX-Serum for 4 hours and washed the cells and incubated in regular media for next four days before adding the breast cancer cells?

Response: Yes, except that the treatment time is 6 hours (see response to Critique 2).

Critique 4: If authors have followed above experimental strategy in comment#3 could authors please clarify the rational for not maintaining the cells in CTX-serum throughout culture period as host-secreted factors are considered as the basis for the inability to recapitulate in vivo findings as mentioned in Figure 2 data.

Response: When we initiated the experiments, we hypothesized that some important host serum factors might have been present at 16 hour. They can set things in motion to change the lung vascular microenvironment, leading to the increase of intravascular cancer cell number in vivo (as we observed). If that is the case, we are likely to detect the effect of CTX-serum by the assay in vitro. If not, a careful time course analysis (collecting serum at different time points after CTX treatment) would be necessary. Since we found differences in the PBS- versus CTX-serum—as assayed by in vitro cell adhesion assay—we followed up on that result. However, we suspect that a careful time course study may show waves of changes in the serum factors at different time points. Those different factors may have different functional impact on the host lung microenvironment to affect cancer cell behavior. The experiment suggested by the reviewer is interesting and would require the change of serum throughout the experiments to mimic the “changing waves of serum factors” in vivo (if that is the case).

Critique 5: Did author perform any studies to examine the difference in maintaining cells with and without CTX serum till the study endpoint on adhesion of cancer cells.

Response: No, we did not. See response to Critique 4 regarding the potential “changing waves of serum factors.”

Critique 6: The authors have stated 16 hr time window was chosen to ensure accumulation of any secreted molecules in circulation in response to CTX. If host derived factors were the reason, the authors should have collected at a later time point such as at 96h to be consistent with their rational on not only the clearance of the chemotherapeutic agent from the system but also to rule out the impact of CTX on signaling cascade? Could authors please provide comment on it.

Response: We did not collect the serum at 96 hours, because the hypothesis we were testing is that, at 16 hours after CTX injection, some important serum factors are already present and can set things in motion to change the vascular microenvironment, leading to increased cancer cell adhesion. See more discussion in our response to Critique 4.

Critique 7: Did authors by any chance examine whether longer treatment of mLECs with 4-OOH could alter breast cancer cells adhesion … ?

Response: No, we did not. This is because the host cells are not exposed to active CTX for a long period. As described in our response to Critique 1, the half-life of CTX in mouse is about 3-4 hours. Therefore, the concentration of CTX that the host cells encounter is reducing continuously due to CTX elimination.

Critique 8: Could authors please clarify whether chemotherapeutic agents were administered intraperitoneally or through intravenous injections.

Response: We thank the reviewer for pointing out this mistake and corrected it in the revised manuscript.

Minor Concerns:

The data shown in Figure 1B is representative of data from 3hr post injection of respective breast cancer cell lines? The authors have used 3hrs as the end point for the studies. So could authors please have the arrow before 3 hrs and remove the arrow after 3 hrs to avoid ambiguity of other timepoints.

Response:

(i) Figure 1B is for MVT-1 cells but is pre-treated with different drugs: CTX or PTX. Figure 1C is for a different cell line (Met-1) using CTX.

(ii) We changed all schematics with the arrows ending before the timepoints.

Round 2

Reviewer 2 Report

The authors have sufficiently addressed concerns raised by my previous review. This is an interesting paper which will add to the readership of the journal. To me, the manuscript is now suitable for publication.

Reviewer 3 Report

The authors have addressed the concerns and the manuscript can be considered for publication.